# Melatonin, Clock Genes, and Mammalian Reproduction: What Is the Link?

**DOI:** 10.3390/ijms222413240

**Published:** 2021-12-08

**Authors:** Amnon Brzezinski, Seema Rai, Adyasha Purohit, Seithikurippu R. Pandi-Perumal

**Affiliations:** 1Department of Obstetrics & Gynecology, The Hebrew University-Hadassah Medical Center, Jerusalem 91120, Israel; 2Department of Zoology, Guru Ghasidas Vishwavidayalaya (A Central University), Koni, Bilaspur 495009, India; drseemakamlesh@gmail.com (S.R.); purohit@gmail.com (A.P.); 3Somnogen Canada Inc., College Street, Toronto, ON M6H 1C5, Canada; pandiperumal@gmail.com; 4Saveetha Institute of Medical and Technical Sciences, Saveetha Medical College and Hospitals, Saveetha University, Chennai 600077, India

**Keywords:** melatonin, clock genes, mammals, polycystic ovarian syndrome, PCOS, reproduction, circadian rhythms

## Abstract

Physiological processes and behaviors in many mammals are rhythmic. Recently there has been increasing interest in the role of circadian rhythmicity in the control of reproductive function. The circadian rhythm of the pineal hormone melatonin plays a role in synchronizing the reproductive responses of animals to environmental light conditions. There is some evidence that melatonin may have a role in the biological regulation of circadian rhythms and reproduction in humans. Moreover, circadian rhythms and clock genes appear to be involved in optimal reproductive performance. These rhythms are controlled by an endogenous molecular clock within the suprachiasmatic nucleus (SCN) in the hypothalamus, which is entrained by the light/dark cycle. The SCN synchronizes multiple subsidiary oscillators (clock genes) existing in various tissues throughout the body. The basis for maintaining the circadian rhythm is a molecular clock consisting of transcriptional/translational feedback loops. Circadian rhythms and clock genes appear to be involved in optimal reproductive performance. This mini review summarizes the current knowledge regarding the interrelationships between melatonin and the endogenous molecular clocks and their involvement in reproductive physiology (e.g., ovulation) and pathophysiology (e.g., polycystic ovarian syndrome).

## 1. Melatonin and Reproduction

The circadian rhythm of melatonin synthesis in the pineal gland is involved in the regulation of mammalian (including humans) reproduction [1,2,3,4]. In 1963, Wurtman et al. [5] first reported that melatonin administration reduces the weight of the ovaries of female rats. Since then, abundant evidence has been reported that the pineal gland, via melatonin, affects reproductive performance in a wide variety of species. In most animals (but not all), melatonin has an antigonadotrophic effect [6,7,8]. Melatonin exerts its actions mostly by acting through membrane-bound receptors, MT1 and MT2 [9]. These receptors belong to the super-family of G-protein coupled receptors containing typical seven transmembrane domains [10,11]. Melatonin might also have a local effect on the ovary. Substantial concentrations of melatonin were found in ovarian follicular fluids [12], and melatonin receptors were reported in ovarian granulosa cells [13]. The seasonal changes in the number of hours per day that melatonin is secreted mediate the temporal coupling of reproductive activity to seasonal changes in day-length [8].

The circulating plasma levels of melatonin are relatively high in childhood and decrease significantly during puberty. This observation stimulated a search for a role for the pineal gland and melatonin in the timing of the initiation of puberty. However, conflicting reports appeared in the literature regarding the role of melatonin in the human process of puberty. Some researchers found higher plasma melatonin levels associated with prepubertal and delayed pubertal conditions [14,15] and inversely lower levels of melatonin after puberty or in cases of precocious puberty [16,17]. Others found no significant differences between normal and disordered puberty [18,19,20]. These discrepancies have led to skepticism among clinicians regarding the importance of melatonin in normal pubertal development. In both young males and females, the puberty-related decline of high childhood melatonin levels has been correlated more to advancing Tanner stages than to chronological age [21]; however, no clear causative basis for this relationship has been established for humans.

Regarding pregnancy and parturition, melatonin serves as a signal from the maternal circulation to the fetus [22]. Melatonin crosses the placenta [23] and can bind to melatonin receptors in numerous fetal tissues [24]. Melatonin receptors (MT1 and MT2) have been detected in human placentae and shown to be expressed throughout pregnancy [25]. It has been reported that high in-vitro melatonin administration significantly elevated hCG release by human trophoblast cells [25,26]. More recently, a high level of in vitro melatonin was found to protect trophoblast cells against hypoxia-induced inflammation [27]. Hobson et al. [28] reported modest improvements in the duration of pregnancy in a small study of women with preeclampsia who received 10 mg oral melatonin three times a day. These preliminary results may suggest new therapeutic possibilities to improve clinical outcomes for women with preeclampsia.

There is thus not enough information on the potential association between reproductive clinical syndromes and melatonin excess or depletion. Clinical experience related to this issue has yielded inconclusive and sometimes conflicting results [4,29]; therefore, further clinical studies are warranted.

## 2. Molecular Clock Genes

In mammals, the main biological clock is located within the suprachiasmatic nucleus (SCN) in the hypothalamus. It receives environmental light/dark information and synchronizes circadian rhythms [30,31,32,33]. Molecular components of the circadian clock are a set of clock genes that involve intracellular transcriptional/translational feedback loops. These loops have negative (*Per1–3*, *Cry1&2*) and positive limbs (*Bmal1* and *clock*). *Bmal-1* and *clock* are the positive transcriptional loop that binds to the promoter region. The binding of *Bmal-1* and *clock* activates the transcription of the *period (per 1–3)* gene and the *cryptochrome (cry)* gene. As a negative transcriptional loop, *PER* and *CRY* proteins heterodimerize to repress the transcription of *Bmal-1* and *clock*. *PER* and *CRY* proteins dissociate from each other and activate *Bmal-1* and *clock* [34,35].

Mammalian clock gene studies have revealed molecular clocks in many brain regions (such as the pineal gland) [36]. Clock genes also exist in peripheral organs such as the liver, lung, kidney, skeletal muscle, GI tract, and female reproductive organs such as the ovaries [37].

## 3. Clock Genes and Reproduction

For regular reproductive physiology, there must be the regular secretion of gonadotropin and reproductive hormones. The timing of hormone secretions depends on the clock genes present in the ovary [38,39,40]. The presence of clock mRNA (*Bmal-1* and per) has been observed in the antral follicle, luteinized granulosa cells which might be entrained by SCN to maintain the coordination between the environmental clock and biological clock. Failure in this coordination may hamper the timing of the secretion of the reproductive hormones leading to various reproductive disorders [40,41].

We (AB) recently reported [37] that the circadian genes *CRY1*, *CRY2*, *PER1*, *PER2*, *CLOCK*, *ARNTL*, *ARNTL2*, and *NPAS2*, are all expressed in cultured human luteinized granulosa cells. Among these genes, there was a general trend of decreased expression in cells derived from older women; however, this was only determined to be of statistical significance for the *PER1* and *CLOCK* genes. This decline may partially explain the decreased steroidogenesis and impaired fertility associated with reproductive aging.

There are other reports suggest interactions between clock genes and reproduction (Table 1). For example, a report detailing that Bmal1 interference impairs hormone synthesis and promotes apoptosis in porcine granulosa cells [42], or the recent finding that LH surge induces a change in gene expression within the GCs of the preovulatory follicle [43]. It has also been reported that there are temporal effects of human chorionic gonadotropin on the expression of the circadian genes and steroidogenesis-related genes in human luteinized granulosa cells [44]. Others [45] reported that the clock genes *BMAL1* and *SIRT1* are necessary for follicle growth and estrogen biosynthesis. 

## 4. Melatonin—Clock Genes Interactions

Previous studies indicated that the disruption of melatonin secretion or clock gene expression might promote inflammation or cancer development and, therefore, might be associated with various diseases (e.g., Alzheimer’s disease, liver disease) [56,57]. Recently there has been increasing interest in the possible role of the distraction of melatonin secretion and clock gene expression in reproductive pathologies.

Sympathetic innervation of the mammalian pineal gland is activated at darkness by a multi-synaptic pathway from the SCN (Figure 1) to release noradrenaline, which acts on the adrenergic receptors of the pineal cells to trigger the cAMP signaling pathway [58] and thus, leads to the activation of melatonin synthesis. Similarly, clock gene *Per1* is rhythmically expressed in the pineal under the same noradrenergic control from the SCN as the one that regulates melatonin synthesis [59,60]. Thus, the molecular clock of the pineal (at least in rodents) seems to be synchronized by the central clock in the SCN. Although the role of the pineal molecular clock has not been fully elucidated, its involvement in the gated expression of N-acetyltransferase, the key enzyme of melatonin biosynthesis, has been proposed [61]. Interestingly, clock gene *Per1* is rhythmically expressed in the rodent pineal under the same noradrenergic control from the SCN as the one that regulates melatonin synthesis [62,63,64]. Thus, it seems that the molecular clock of the pineal gland is synchronized with the central clock in the SCN [65] (Figure 1).

## 5. Melatonin, Clock Genes, and Gonadotropins Disruption: The Case of PCOS

Polycystic ovarian syndrome (PCOS) is a common gynecological, endocrine disorder (8.7 to 17.8% of reproductive age women). It is most commonly observed in young women [66,67]. The main features of the syndrome are irregular menses (oligomenorrhea), anovulation, obesity, hirsutism, impaired glucose tolerance, and multiple ovarian cysts. The syndrome was first described by Stein and Leventhal in 1935 [68] and further characterized by others. In 2003, a consensus workshop sponsored by the European society of human reproduction and embryology (ESHRE) in Rotterdam indicated PCOS to be present if any two of the criteria (i.e., anovulation, obesity, hirsutism, impaired glucose tolerance, and multiple ovarian cysts) are met [69].

The mechanisms underlying the development of PCOS are still uncertain. It is well known that there is a strong association between hyperandrogenism and oligo/amenorrhea, anovulation, and hirsutism [70,71]. Nevertheless, the exact mechanism causing this association is not clear. What further complicates this issue is the recent report that PCOS may be associated with genetic/epigenetic factors [72].

It has been established that a correlation exists between female reproduction and the circadian clock. This association suggests that there might be a link between clock genes and female reproductive disorders. There are several pieces of evidence of disrupted circadian rhythm causing various reproductive impairments [41]. There are indications that direct crosstalk exists between clock genes mechanisms and PCOS conditions. For example, a changing lifestyle and night shift work are associated with an increased risk of developing PCOS [73].

There are increasing reports regarding the presence of various clock genes in the ovary, and that the disruption of these genes is associated with impaired ovarian function [37,50,74,75]. Reduced Bmal1 has been noticed by Li et al. in constant darkness, causing PCOS. It also has been reported that during polycystic conditions, per1 and per2 levels decrease [53]. Decreased expression of the clock gene ARNT-like protein 1 reportedly mediates the contribution of hyperandrogenism to insulin resistance in PCOS [75].

Per1 and per2 clock genes are associated with successful ovulation as both are responsible for optimum LH surge [74]. During PCOS, the ratio of LH/FSH increases due to excessive secretion of LH and the role of per1 and per2 becomes hampered [49,74]. Similarly, insulin resistance and hyperandrogenism, two of the main features of PCOS [76], are associated with decreased Bmal1 expression [53,77]. Bmal1 interference impairs hormone synthesis and promotes apoptosis in porcine granulosa cells [42].

Disrupted circadian rhythms are very common due to changing lifestyles and irregular sleeping patterns [78]. Therefore, the abnormal secretion of gonadotropins and reproductive hormones due to impaired clock genes and disrupted melatonin secretion may explain some of the basic pathophysiologies of PCOS.

It has been reported long ago that melatonin rhythms are disrupted in night shift workers [79]. Similar disruption was found in women with PCOS [80,81,82,83].

Most recently, in a combined humans/animals study, the association of circadian rhythm disruption with PCOS was investigated [84]. They found that there was a significant correlation between night shift work and PCOS. In addition, PCOS-model rats presented distinct differences in the circadian variation of corticotropin-releasing hormone, adrenocorticotropic hormone, prolactin, and a 4-h phase delay in thyrotropic hormone levels.

As critical endogenous messenger melatonin may have a role in restoring circadian rhythmicity in disrupted ovarian cells of PCOS [85,86]. Acting as a rhythm synchronizer, melatonin has been shown to regulate central and peripheral clock genes [87,88,89]. Melatonin might also have a direct local effect in the ovary as significant amounts of the hormone were detected in human preovulatory follicular fluid [12] and melatonin receptors were detected on human granulosa cells membranes [13]. Similarly, the two enzymes (AANAT; Arylalkylamine N-acetyl-transferase, and HIOMT; Hydroxy indole-O-methyltransferase), that play a major role in the synthesis of melatonin have been detected in ovarian tissue [90].

## 6. Conclusions

The data presented above indicate that the circadian rhythm of the pineal hormone melatonin and clock genes play a role in synchronizing the reproductive responses of mammals to environmental light conditions and circadian rhythmicity. Both melatonin and clock genes are present in ovarian tissues, and there are indications suggestive of interactions between clock genes, melatonin, and reproduction. Currently, the balance of evidence from in vitro and in vivo studies suggests that rhythmic pineal melatonin production and clock gene expression are disrupted or attenuated in pathological reproductive conditions such as anovulation and PCOS. The exact mechanism by which melatonin and clock genes affect mammalian reproductive processes is still to be elucidated.

## Figures and Tables

**Figure 1 ijms-22-13240-f001:**
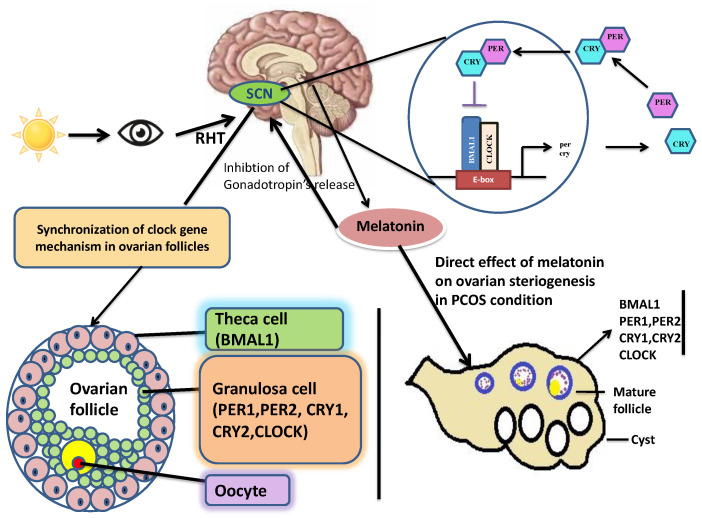
Possible reproductive pathways of melatonin and clock genes. Melatonin and clock genes are involved in optimal reproductive performance. Their rhythms are controlled by an endogenous molecular clock within the suprachiasmatic nucleus in the hypothalamus, which is entrained by the light/dark cycle. Both melatonin and clock genes exist in ovarian granulosa cells. Disruption of melatonin and clock genes rhythms interferes with ovarian steroidogenesis and reproductive function. SCN = suprachiasmatic nucleus, RHT = retinal-hypothalamic tract.

**Table 1 ijms-22-13240-t001:** Summary of clock genes’ location and the proposed mechanisms of the effects on mammalian reproduction.

Clock Genes	Location	Proposed Function	References
PER1	SCN, premature ovarian follicle	Participate in the coordination of GnRH and LH surge.	[46] Zheng et al., 2019
		Melatonin helps in the regulation of the expression of clock genes at neural and peripheral tissue levels.	[47] Coelho et al., 2015;
		Knockdown of Per1 or Per2 hampers fertility by the disruption of the estrus cycle.	[48] Toffol et al., 2016
		Per1 plays a role in sustainable pregnancy by stimulating progesterone secretion.	[46] Zhang et al., 2019
PER2	SCN, ovarian corpora leutium, granulosa cell, and oviduct	Knockdown of PER2 leads to the flattened diurnal oscillation of all the core clock genes and to disorganized human endometrial stromal cells.	[45] Zhang et al., 2019
		Per2 influence granulosa cell functions, including cell proliferation, steroid production, and LH receptor expression, (which are associated with the follicular recruitment and selection of follicles.	[49] Nagao et al., 2019
		Mutations in Per2 genes reduces the number of ovarian follicles and leads to a adecrease in the fertility process.	[50] Pan et al., 2020
CRY1/CRY2	SCN, leutinized granulosa cell	An age-related decreased expression of these clock genes may partially explain the decreased fertility and steroidogenesis of older women.	[37] Brzezinski et al., 2018
BMAL1	SCN, ovary (antral follicles, corpora leutium), oviduct, GnRH neuron	Bmal1 expression in progesterone induction from leuteinized granulosa cells and maintenance of the postovulatory progesterone surge.	[49] Nagao et al., 2019
		Expression of BMAL1 in the GnRH neuron determines the timing of secretion of GnRH.	[51] Sen and Sellix, 2016
		Brain and muscle ARNT-like protein 1 (BMAL1) is necessary for fertility and is essential for follicle growth and steroidogenesis.	[45] Zhang et al., 2016
		Minor changes in Bmal1 can alter the timing of LH surge.	[41] Sen and Hoffmann 2020
		Deletion of BMAL1 in ovarian theca cells disrupts the ovulation.	[52] Mereness et al., 2016
		Bmal1 plays a role in the molecular clock of ovarian steroidogenic cells, the production of progesterone, and other aspects of female reproduction.	[50] Pan et al., 2020
CLOCK	SCN, ovary (late antral follicle)	Knockdown of CLOCK expression in the ovary leads to a significant reduction in litter size and oocyte release.	[50] Pan et al., 2020
		The heterodimer of BMAL1 and CLOCK determines the timing of secretion of the steroid hormone.	[53] Li et al., 2020
		Rhythms of *lhcgr* expression are directly regulated by the BMAL1:CLOCK enhancer complex binding to E-box motifs in the *lhcgr* promoter.	[54] Sellix, 2015
		Defective CLOCK protein enhances the rate of anovulation and pregnancy failure.	[55] Sciarra et al., 2020

## Data Availability

No datasets were generated or analyzed during the current study.

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
