# Peer review of "Melatonin, Clock Genes, and Mammalian Reproduction: What Is the Link?"

_ijms, 2021, doi:10.3390/ijms222413240_

Round 1

Reviewer 1 Report

The present review reviews the role of the circadian system in female reproduction. It is quite clear, goal-oriented written. However, there are a lot of space and formatting errors- more than I want to correct individually. Please check the text for these mistakes.

Page 2 Line 87: “these hormone secretions” sound awful. Rather use just hormone secretion.

Page 4, Line 124: The AANAT is a key enzyme of melatonin synthesis. It should not be called the “rhythm generating enzyme”, since it leads not always to a rhythmic expression.

Author Response

Point 1: The present review reviews the role of the circadian system in female reproduction. It is quite clear, goal-oriented written

Response 1: we thank the reviewer for her/his positive remark and appreciate his/her constructive suggestions.

Point 2: There are a lot of space and formatting errors- more than I want to correct individually. Please check the text for these mistakes.

Response 2: We found these multiple errors and corrected them. (Please see the revised version)

Point 3: “these hormone secretions” sound awful. Rather use just hormone secretion.

Response 3: The requested change was made, thank you.

Point 4: The AANAT is a key enzyme of melatonin synthesis. It should not be called the “rhythm generating enzyme”, since it leads not always to a rhythmic expression.

Response 4: We absolutely agree and we rephrased this sentence.

Reviewer 2 Report

The manuscript by Brezinski and colleagues entitled, "Melatonin, Clock genes, and Mammalian reproduction: What is the link?," is a brief review on the potential association of PCOS with dysregulated circadian pineal melatonin production and clock gene expression. This is a timely minireview as the field of disrupted circadian rhythms in human health and disease is an area of keen interest to researchers and clinicians in multiple disciplines. I have no major concerns with the manuscript, however there are some minor issues that need attention to improve the readability of the manuscript.

Throughout the manuscript and figure legends there are many typographical errors that need to be corrected.

Lines 119-21 and 125-127 make the exact same point using the same phrasing, but using different references. Please corrrect/revise. 

Author Response

Point 1: This is a timely minireview as the field of disrupted circadian rhythms in human health and disease is an area of keen interest to researchers and clinicians in multiple disciplines. I have no major concerns with the manuscript,

Response 1: We thank the reviewer for this positive comment. We join his view that disrupted circadian rhythms is becoming an important issue in various areas of research.

Point 2: Throughout the manuscript and figure legends there are many typographical errors that need to be corrected.

 Response 2: We found these typographical errors and corrected them. (Please see the attached revised version).

 Point 3: Lines 119-21 and 125-127 make the exact same point using the same phrasing, but using different references. Please corrrect/revise. 

Response 3: Lines 119-121 describes the possible role of the distraction of melatonin secretion and clock genes expression in reproductive pathologies, while lines 125-127 describe the mechanism by which adrenergic receptors of the pineal cells trigger the cAMP signalling pathway and thus, leads to the activation of melatonin synthesis. So frankly we do not see any duplication.